# Clean-Up Workers (Deluxe Series): The Embodiment of Waste Values and Aesthetics

## Gayle Matthias

Arts, Design and Media, Arts University Plymouth, Plymouth PL4 8AT, UK; gmatthias@aup.ac.uk

**Abstract:** Written from the perspective of practice-led research, this reflective case study rationalises and charts the production of 'Clean-Up Workers (Deluxe Series)'—mixed media sculptures that embody notions of waste aesthetic, value and abjection. Integrating discourses surrounding waste theory and using the sink and plug as a metaphor to discuss Lacan's theory of the *objet petit a*, the paper is presented as an autobiographical waste narrative. Production of a series of anatomical vacuum cleaners made from re-appropriated artwork found waste materials in the form of 'pre-owned objects or materials' and 'by-products' of a creative practice', sit alongside crafted luxurious glass objects and speak of corporeal ageing, dysfunction and the domestic realm. Discarded objects take the form of car parts found in the non-places of the gutter. Through assemblage, these unique items' 'use-time' is recontextualised and elevated as art objects viewed within a gallery arena.

**Keywords:** narrative; materiality; corporeal; waste; non place; value; readymade; abjection; sculpture

## 1. Introduction

'Much of modern art is a testimony to our need to pick through the rubble of the past, choosing here and there a cloak of symbols which still fits, but mostly aimed at extracting meaning from personal experience'. (Hollis 1993, p. 86)

This essay documents practice-led research that addresses ideas around the waste that has been produced as a consequence of mixed media sculptures that embody waste aesthetics and waste associations. I will illustrate my concepts by providing a reflective case study of the origins of a series of works entitled Clean-Up Workers (Deluxe Series).

My mixed media sculpture represents the body as a metaphor for waste/ageing/ dysfunction on a personal and geographical scale. Abstract anatomical assemblages combine cast, blown and sheet glass with an array of upcycled discarded objects and worn cleaning materials.

'The inclusion of waste materials in sculpture is a 20th century phenomenon, emerging through the traditions of assemblage associated with Cubism, Dadaism, Surrealism and Arte Povera' (Whiteley 2011, p. 8). I have been incorporating waste objects and materials in my work since 2007 and am one of a small number of artists working with glass that combines quotidian materials and objects in this manner. Examples of other contemporary glass artists working in this manner are Emma Woffenden, Matt Durran and Helen Pailing. Abjection and appropriation are explored through the collection and utilisation of throw-away materials and objects, material hierarchy and the re-appropriation of old artwork that I deemed to be ill-conceived. The use of waste as a by-product of my creative practice has been elevated and incorporated into the finished work. This has resulted in the development of a material economy in my practice. I consider the aesthetics of waste materials and objects as an expansion of my visual language. Collecting these materials has enhanced my awareness of what I identify as finished work or waste work/materials. Storing waste for reuse opens discussions about value and consumption within my practice. I am not suggesting that sustainability is a driver for my practice, nor that all objects should be valued and kept as there are consequences of 'not letting go', which leads to dysfunction.



But I am interested in what society and artists throw away and why, what and how things are classified as waste, including bodily waste.

The sites of my waste acquisitions, from the roadside and salvage yards, deal with notions of non-spaces and collective non-responsibility for human disposal of unwanted items, but also stories of carelessness, rejection, indifference and disdain. These places where waste accumulates are signs of dysfunction and neglect. The classification of people who are associated with waste, from domestic to municipal handlers of waste, is of interest in regard to perceived low social status and value.

A key sign I deploy is sanitary ware, particularly the sink and plug combination. I refer to the sink as a pristine barrier or gateway between sanitisation and the hidden depository and processing of waste and how this is a metaphor for a postmodern society's inability to acknowledge and discuss waste, especially corporeal waste (though recent headlines about the frequency of sewage entering the waterways have highlighted the problem of waste management in the UK).

My research stems from autobiographical themes to do with the impact and decline of the former heavy industries located in Stoke-on-Trent (The Potteries), my birthplace. The Potteries was home to pioneering sanitary ware production of the Victorian British Empire but is now mainly a heritage site fallen into disrepair, in desperate need of levelling up. Former collieries connect in my mind with my father's car body repair garage, which was sited in a former miners' baths. The paper S-O-T Body Repairs: Narrative Pursuits (Matthias 2019) reflects in detail on this underpinning theme to my practice. Therefore, I turn my attention to the narrative of an unfulfilled life of domestic chores and isolation that was experienced by my mother and other female family members of that silent generation.

## 2. Results

The concluded artwork discussed in this paper pays homage to domestic cleaners through the production of Clean-Up Workers (Deluxe Series): imagined shabby, dysfunctional anatomical vacuum cleaners which unite domestic cleaning materials, by-products of my creative production, including re-appropriated work made from objet trouvé (sink) and waste objects from car parts and glass in complex compositions that capture complex narratives. Theoretical research into waste and discard theories combined with autobiographical narratives was applied to the design development and ethos of the final sculptures. These mixed media sculptures are located on the periphery of glass art; the work could be perceived to challenge some preconceptions of traditional glass art practice, which has a tendency to focus on technical expertise and material singularity. I advocate an artistic scavenger mentality in order to both economise and expand both conceptually and materially. To date, the work has been selected for two prestigious glass exhibitions, resulting in the acquisition of Clean-Up Workers II (Deluxe Series) by the European Museum for Modern Glass, Coburg, for their permanent collection.

## 3. Discussion

### 3.1. Material Value

Trash has been a central feature in a diverse range of art practices throughout the twentieth century. 'The objet trouvé was prized and fetishised by Surrealists but 'junk art' is primarily associated with the idiom of assemblage—a set of object-based practices which emerged in the mid 1950's and culminated in the seminal exhibition The Art of Assemblage in New York in 1961' (Whiteley 2011, p. 8). I align with the methodologies of assemblage and bricolage; however, I distinguish my practice in that I combine low-value materials and objects with high-end materials and processes that I create from scratch; these components have origins in craft production, such as glass casting. The material culture and associated aesthetics of glass make this material relevant to my visual language. It is not a material that I use in all my work, and the amount that is in a composition can be minimal. However, moulding found objects mainly from under the bonnet of a car allows modification through translation and combination of waxes to create unique glass

casts that have corporeal associations of body dislocation and capture material optics and material value.

Glass, or rather kiln-formed glass, is as highly valued in contemporary society as it was at its birth in Mesopotamia from 2000–3000 BC. Glass, being a synthetic material, a precursor to plastics given its mouldable nature, was originally used to imitate precious and semi-precious stones such as agate, opal and lapis lazuli (Cummings 1997; Macfarlane and Martin 2003). I sometimes exploit this material mimicry by casting with opaque frits.

According to Cummings, 'the evolution from industrial to post-industrial society has seen the rebirth of kiln-formed glass, after almost 2000 years, in a direct way. The main processes and techniques, the types of objects made by them, their relationship to society in terms of value, function and philosophy are remarkably unchanged and therefore of extreme relevance to our own era and practice' (Cummings 1997, p. 24)

Kiln-formed glass was, and still is, due to the nature of the laborious hand-crafted procedures, costly to make, and the resulting outputs tend to be produced on a small scale and have limited functionality, catering for 'an elite clientele who valued the luxurious exclusive goods produced by the system' (Cummings 1997). I also outsource studio-blown glass components, which possess a similar value to kiln-formed glass, given that they are one-off, individually blown items. The Romans refined and capitalised on glassblowing. It became transformational and ubiquitous because of their efficient, large-scale technological advancements that resulted in cheaper functional glass production that usurped kiln-formed glass production and culminated in a long hiatus in kiln-formed glass due to the loss of technical knowledge. It also pushed kiln-formed glass into the decorative arena.

Exploiting the tension between this opulent material, glass—with broken found objects and by-products or waste materials—therefore raises questions about material value and worth. Does the inclusion of glass items elevate the hierarchy of junk or vice versa? Emma Woffenden's mixed media sculptures address similar questions about material value. She makes assemblages from manufactured glass bottles, plaster and high-density modelling foam, all of which sit uncomfortably alongside crafted glass objects in confrontational figurative compositions. When I dismantle Clean-Up Workers (Deluxe Series), the individual components resemble random items to be skipped, though each item is unique in its wear and tear and, therefore, irreplaceable. So, the authentic waste item is combined in an act of bricolage with the luxurious handmade. According to Russian artist Ilya Kabakov, who presented garbage installations and referred to his practice as 'biographical garbage', 'the phenomenon consists of the fact that garbage in art today is presented as treasure—and in essence this is the case' (Lange-Berndt 2015, p. 111).

### 3.1.1. Categorisation of Waste

Objects and materials become waste for different reasons; objects and materials possess a hierarchy of value that instigates human care for objects or materials, or not. I perceive that waste falls into two categories: the 'by-product object or material' and the 'owned object or material'.

As 'by-product object or material', in this case, the waste from making artwork, by-products relate to what we throw away (excrete) after consumption (digest) (Scanlan 2005, p. 138). Precious cast glass has always been remelted where possible wax recycled, but I have thrown away an abundance of plaster and other materials over the years. 'In perceiving we are building, taking some cues and rejecting others' in a system of ordering and classification as to what is and isn't appropriate or valuable' (Douglas 2002, p. 45). I was paying attention solely to my alluring material of choice, glass. Since 2016, due to a change in creative narrative and, therefore, material usage, I have saved excess set plaster and repurposed it as symbolic of waste, mining and flesh (see Figure 1). I keep shavings, swarf and dust created from filing, cutting and drilling found objects and reinstate them into the finished work as debris that has been corralled by a cleaning device. In 'Civilisation and its Discontent', Freud identified that besides cleanliness, beauty and order are key requirements of civilisation, 'Squalor of any kind seems to us incompatible with civilization,

and we extend the demand for cleanliness to the human body too' ([Freud 2014](), p. 43). Through my sculptures, I want to address waste aesthetics and aspects of disorder and, hence, the threat presented by waste.

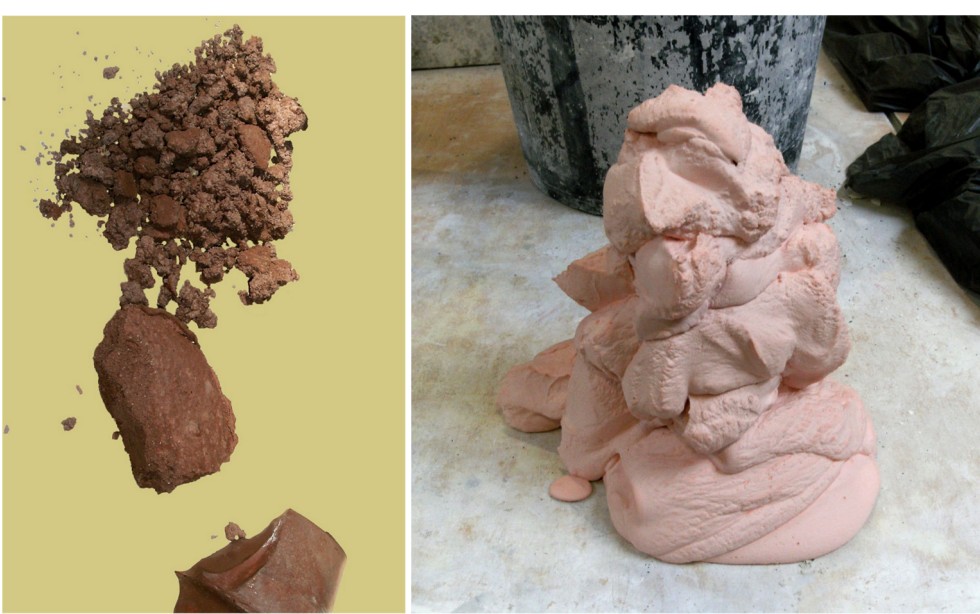

**Figure 1.** By-product objects or materials from a creative practice. 2023

Anthropologist Mary Douglas categorises the different stages of degeneration of waste materials, the uncanny stage of objects/materials becoming disorderly—when they could be broken, spent, rejected, or become by-products. "This is the stage at which they are dangerous: their half-identity still clings to them and the clarity of the scene in which they obtrude is impaired by their presence" ([Douglas 2002](), p. 197). They are disturbing in their ambiguity and potential for pathogenic harm, although this does depend on the nature of the composite. This correlates with philosopher and psychoanalyst Julia Kristeva's theories about the causes of abjection, 'It is thus not lack of cleanliness or health that causes abjection but what disturbs identity, system, order. What does not respect borders, positions, rules. The in-between, the ambiguous, the composite' ([Kristeva 1982](), pp. 2–4). The use of waste and dirt alongside glass is to provoke an emotional response to my work, albeit a negative response, the desire to label and/or dismiss (as ugly).

My use of found objects adheres more to the definition of half identity than the reuse of by-products, which may already lack clear definition and may be seen to be on their way to pulverisation and eventual formlessness that equates to dirt. Formlessness, according to Douglas, presents a reduced threat beyond the disorderly as it suggests both the potential for growth as well as decay. I would argue that this very much depends upon the composition of that 'dirt'. 'Purity and Danger' was first published in 1966, before the awareness of the dangers of microplastics.

Cornelia Parker is an example of a visual artist who uses waste by-products in poignant ways within her sculptures. The large-scale immersive installation War Room (2015), comprising the reclaimed, perforated remnants from the British Legion remembrance poppy (rolls of paper negatives covered the walls and were draped across the ceiling of the Whitworth Gallery. 300,000 holes related to lives lost, the work is about absence rather than presence). Contrast this with her intimate work, The Negative of Words (1996), composed of a simple pile of swarf, the by-product of a silversmith engraving words into silver ([Riddle n.d.]()).

Secondly, the 'owned object or material' could become waste through loss or damage, whether purposeful or not, general wear and tear, or it has become démodé (see Figure 2). This could also relate to sentimentality or nostalgic associations given to an object or material but ultimately relates to impermanence and, therefore, eventual death. William

Viney, in his book 'Waste: A Philosophy of Things', states that 'Objects do not have a certain 'amount' of value until this value is used up'; he argues 'that the value of things is determined by the time of use and waste that we ascribe to them' (Viney 2015, p. 4). He identifies that objects/materials have a 'use time' whereby the 'activity of using things gives shape and order to time and equally, materialises how the time we consign to things brings meaning to use'. Once redundant, these objects or materials have a waste time without a function, and therefore a temporal end' (Viney 2015, p. 8). We can try to extend use-time through repurposing and redefining. I remember my father's old vests were demoted and cut to make cleaning cloths. They would be washed and reused until they could no longer fulfil that lowly service. Before the rise of mass consumerism in the mid-1900s, Western societies used to downgrade objects and materials before they were disposed of to prolong the 'use time'. Now, certain objects such as single-use cleaning wipes are disposed of straight away as if dead; their 'use time' is seconds and their waste time is hundreds of years. Subsidiary or 'supplementary supporting objects or materials', such as quotidian packaging, surplus to requirements, already labelled as disposable, are a subcategory, if not an antithesis, of the owned appreciated object.

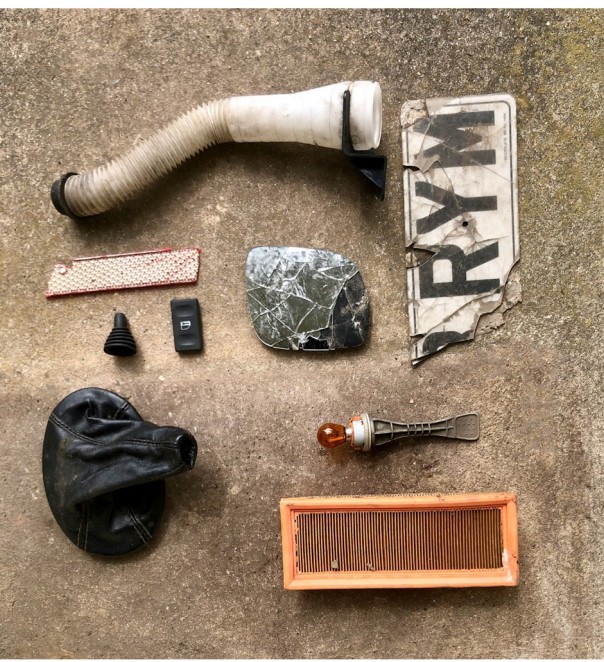

**Figure 2.** Found objects from car bodies. 2023

Assemblage artist Joseph Cornell (1903–1972) collected, classified and boxed an abundance of waste items in storehouses. He then selected and curated them in aesthetic, anthropological grid-like compositions. Cornell's 'banal objects that form the small and overlooked aspects of, one suspects, a distant childhood where the past can be understood as the debris of a life that somehow continues as a ghost of the present' (Scanlan 2005, p. 92). These waste items are personally and visually synthesised and reinvented by the artist as elevated objects of meaning which are worth spending time with. I recognise this obsession, and the objects that I choose have nostalgic meanings that relate to gender expectations and domestic obsessions. I contemplate the sometimes-lonely existence and repetitive chores of a housewife of the 1960s–1970s, 'Her role is to be the home by being at home. Her being at home gives him comfort and allows him to open on the expanse of the world, to build, to create' (Irigaray 1993, p. 130). My artwork is an amalgamation of these two contradictory states of being, the domestic (mother) and the repair garage (father), represented through object assemblage.

Conceptual artist Mark Dion, on the other hand, produces archaeological archives of objects that are excavated from particular locations, such as Tate Thames Dig (1999). The

leveller of the riverbed treats all waste items as equal, and they were displayed as such in large vitrines. The waste narratives are determined by the topography of that section of the river, resulting in random selections. Both artists created cabinets of curiosities but from different starting points.

### 3.1.2. Waste Places

As previously mentioned, my choice of waste items is not arbitrary; it is narrative-driven. Though it is, in some respects, governed by chance encounters, I am actively searching for relevant discarded materials and objects. I am not advocating that all waste is valuable and should be obsessively hoarded—the implications of that are detrimental.

Implements/appliances used to control waste in a domestic arena—scrapers, brushes, and vacuum cleaners—are my current preoccupation. They can be found in back streets by rubbish bins along with the usual fly-tipping standards of broken sofas and DIY remnants. These sorts of abandoned items can be left for weeks as the refuge collectors refuse to dispose of them, but eventually, they are removed from sight. I regularly take photographs of these 'refuse still lifes' (see Figure 3).

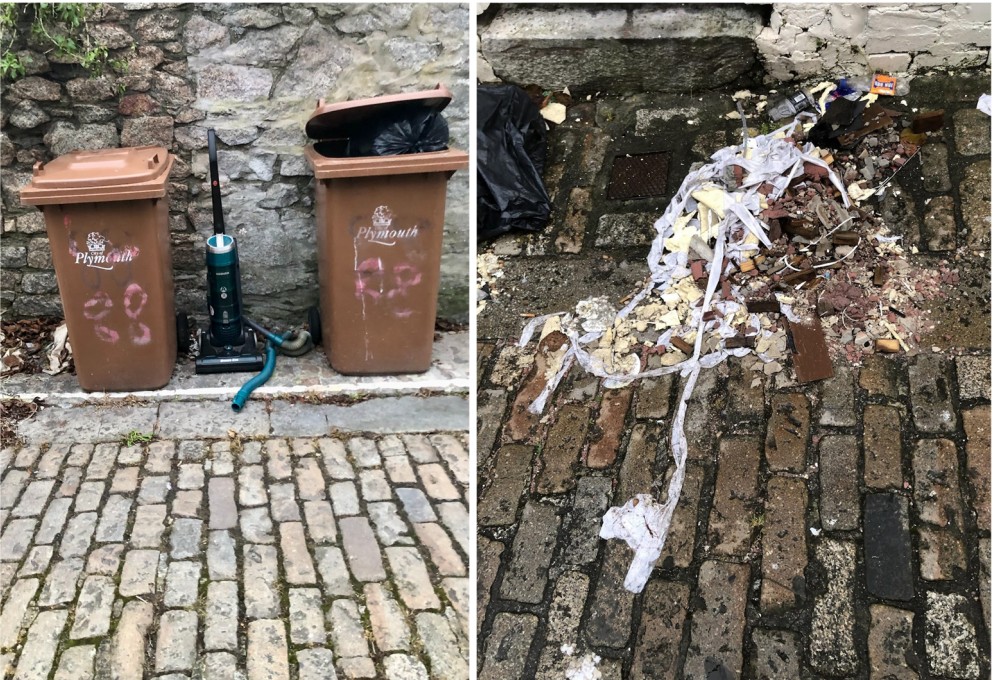

**Figure 3.** Refuse still life photographs, Plymouth, UK. 2023

The other rarer items are discarded car parts that can be found at the side of the road, such as smashed wing mirrors. Transportation such as motorways or train networks are described by sociologist Marc Augé as examples of non-places of supermodernity. 'Super-modernity (which stems simultaneously from the three figures of excess: overabundance of events, spatial overabundance and the individualization of references) naturally finds its full expression in non-places' (Augé 1995, p. 109). I would add to that list the overabundance and consumption of commodities and their resulting waste. Non-places are transient, quite often with the assistance of prescriptive or prohibitive signage to define navigation and behaviour within them. They are seen as possessing negative qualities. 'As anthropological places create the organically social, so non-places create solitary contractuality.' (Augé 1995, p. 94).

> 'If a place can be defined as relational, historical and concerned with identity, then a space which cannot be defined as relational, or historical, or concerned with identity will be a non-place'. (Augé 1995, p. 77)

They are, therefore, places that can be abused through vandalism, careless dropping of waste, or places where waste accumulates (see Figure 4). Because these places appear to belong to everyone, they belong to no one, they are, in some instances, inaccessible. Road verges or central medians are places where waste materials and objects come to rest.

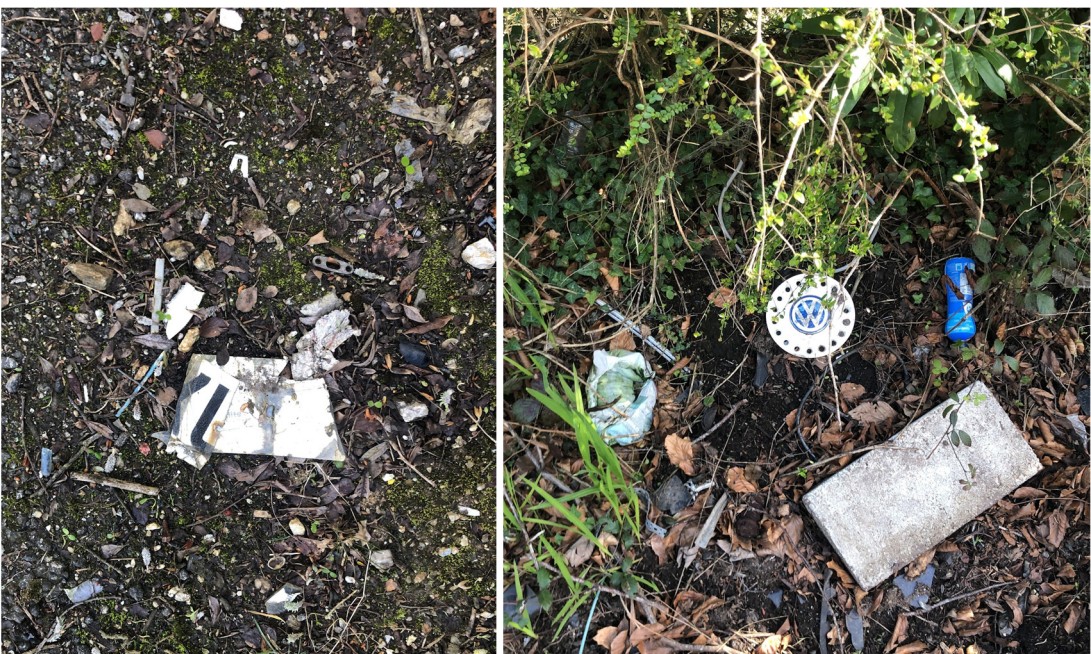

**Figure 4.** Images of non-places and found objects, Orchards Scrap Yard Car Park, Cornwall, UK.

Academics, authors and installation artists Dr Bob Whalley and Dr Lee Miller, in a shared PhD authorship and performative practice, collected around 30 piss bottles from the hard shoulder of the M6 motorway and curated them in their garden shed. Similarly, photographer Ricky Adam in his book M1, between 2015–2017, photographed piss bottles found at various service stations on the M1 motorway. These uncanny items of convenience provoke disgust because of their association with human waste and the way that they are carelessly disposed of, defiling a non-place through bodily pollution that is preserved in plastic and not absorbed into the land. But also because the piss could be misconstrued as a cruel joke. Cast glass can capture the qualities of bodily fluids, and I reference this in my work.

*3.2. Methods and Materials*

The Sink: Bodily Waste and Abjection

A prevailing iconic sign in my work that emerged in 2007 and the catalyst for a series of works that has spanned sixteen years is my use of the sink plughole as an iconic sign that signifies a portal between the underworld/internal workings and outer visible sink. The act of plugging the sink so that it could be used to contain water in order to cleanse the body, also blocking the view of the plumbing beneath, could possibly be an act of denial of the existence of the other. The contrast between the clean, sanitised bowl and the grimy microcosm of the intestinal plumbing pipes, the underworld of anaerobic bacteria, an environment that thrived on limited light and oxygen, human spit, sweat, skin, hair and debris, is analogous to the subterranean landscape of mining. I would also align my practice with 'Abject Art', a term first used by Kristeva in the 1990's. Abject Art has strong associations with feminist contexts; the functions of the female body, in particular throughout history, have been depicted as being impure and, therefore, a threat to society. Such artworks address subjects or use materials that intentionally trigger disgust

or repulsion in the audience (e.g., Judy Chicago, Red Flag, 1971). I see the sink as possessing anatomical female qualities and my use of dirt, debris and waste as being repulsive.

I responded to the plug as an emotional device to suppress and contain internal revulsive feelings that occasionally seep to the surface or cause subsidence as a way of capping or controlling the body's metabolism and psyche.

According to philosopher and feminist theorist Elizabeth Gross, 'all sexual organs and erotogenic zones, Lacan claims, are structured in the form of a 'rim', which is the space between the inside and outside of the body. These corporeal sites provide a boundary or threshold between what is inside the body, thus part of the subject, and what is outside the body, and thus the object for the subject. This boundary must be traversed by the incorporation and/or expulsion of erotic objects. Objects are, in this sense, neither fully contained within the subject's body nor ever entirely expelled from it' (Gross 1990, p. 88).

In general, the skin could be viewed in this way, but the rim or orifice is a more specific/accessible opening. This hypothesis seems to substantiate my sink plug and plughole analogy to the human anatomy and psyche. Douglas states that 'Any structure of ideas is vulnerable at its margins. We should expect the orifices of the body to symbolise its specially vulnerable points. Matter issuing from them is marginal stuff of the most obvious kind.' (Douglas 2002, p. 150).

In discussing the work of Kristeva, Gross identifies that bodily waste

'provoke cultural and individual horror and disgust, symptomatic of our cultural inability to accept the body's materiality, it's limits, its 'natural' cycles and its mortality. Faeces, for example, in signifying that the opposition between the clean and the unclean draws on the distinction between the body's inside and its outside. Inside the body, it is the condition of the body's ability to regenerate itself; as expelled and external it is unclean, filthy. The subject is implicated in this waste, for it can never be definitely and permanently externalised: it *is* the subject; it cannot be completely expelled'. (Gross 1990, p. 91)

Corporeal margins and what they reveal or excrete can be amplified onto social and cultural structures which possess similar thresholds and excesses.

The sink plug directed me to look more closely at the sink itself. In the early 15th century, the sink used to be the "cesspool, pit for reception of wastewater or sewage". A figurative sense of a "place where corruption and vice abound," is from the 1520s, and the sense of a "shallow basin (especially in a kitchen) with a drainpipe for carrying off dirty water" was first recorded in the 1560s (Harper 2023). The term 'behavioural sink', introduced by Ethologist John B Calhoun, was deliberately used to describe the nihilistic behaviour of rats as a result of overcrowding experiments (Wiles n.d.). So, the notion of 'sink' is connected with filth and disorder but has transformed from a place of filth to a sanitised gateway to the world of filth below.

This heightened aspect of revulsion and denial of bodily waste and waste in all its forms I judge to be a modern phenomenon, one that falls into alignment with improved sanitation and the understanding of the microscopic world. 'Indeed, we are not surprised if someone actually proposes the use of soap as a criterion of civilisation. Much the same is true of order, which, like cleanliness, relates wholly to the work of man' (Freud 2014, p. 43). The purpose of all this work is to protect against the forces of nature. The greater the protection, the more civilised we become and the greater the requirement to hide certain civic operations. Science fiction links 'technological advancement and its perceived 'orderliness' to the cold sterility of a clean, waste free vista' (Thill 2015, p. 44). I view the changing role of the sink from an open cesspit with exposed rot and stench to a sanitised white-glazed vessel with shiny faucets, a barred underworld contained within discrete pipes and traps, as a symbol of society's relationship with corporeal waste and mortality.

Italio Calvino, in his essay La Poubelle Agréée, compares taking out the rubbish as 'a rite of purification' separating himself from a part that was once his, analogous to that of defecation, the sensation, at least for a moment, that his body contains nothing but himself, that he is still a producer of detritus and not detritus himself (Calvino 1994, pp. 103–4).

Artists such as Mona Hatoum force the viewer to engage with such bodily detritus and internal workings. As a prime example of Abject Art, Hatoum collected 'nail parings, pubic hair, bits of skin and mixed them with pulp and bodily fluids to make paper, a kind of recollecting of the body's dispersal' (Hatoum 2005, p. 298). Her renowned work, Corps Étranger (1994), followed the endoscopic camera as it travelled down her oesophagus into her stomach, allowing the audience internal access to her body via an enlarged projected portal on the floor, which encouraged the act of looking downwards and inwards or perhaps into an underworld.

With 'Sink and Plug' (see Figure 5), I wanted to isolate and draw the audience's attention to the plughole portal by lining the receptacle of an old sink with roughly cut sheet glass. The glass served multiple purposes: as a signifier of danger, an acknowledgement of the act of plugging and an exaggerator of the internal organ-like quality of the sink bowl, now possessing a striated ridged muscular lining of glass. However, the sheet glass, when assembled, no longer possessed the threat of the individual shard; placing the plug in the plughole would not result in an anxious, raised heartbeat as anticipated.

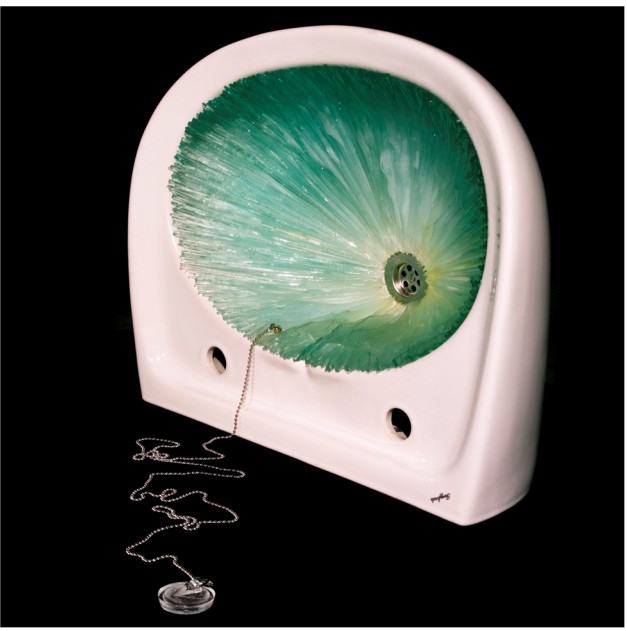

**Figure 5.** Matthias. 2007. Sink and Plug, 43 cm H, 46.5 cm L, 15 cm W. Found object, sheet glass and cast glass.

The audience also perceived the sink to be a comment on Duchamp's controversial and seminal 'Fountain' (1917) (an act, a concept, that transformed the landscape of modern art via the displacement and elevation of ordinary sanitary ware into the space of the art gallery), or Gober or Levain's recontextualisation of Duchamp's work, and this was not my intention. As a consequence, I realised that the sign of the sink was too dominant, and I was, in fact, intentionally contradicting Duchamp's criteria for selecting his ready-mades, 'a reaction of visual indifference with at the same time a total absence of good or bad taste . . . in fact a complete anesthesia' (Cited in Duchamp and Sanouillet 1975, p. 14). My choice was not aesthetically or emotionally neutral. 'Sink and plug' was ill conceived/misleading as the sheet glass, though transparent and on edge, had a turquoise enhancement suggestive of water; the glass, therefore, lacked the desired threat and could also be misinterpreted on several levels. I, therefore, had to decide how to actively break these associations without losing the sink. How many sink fragments were required in a composition to still speak of sink? I discovered not many.

I began to smash old toilets and sinks in order to respond to the revealed slip-cast cavities through 3D scanning, conventional moulding, sheet glass extensions, or the addition of props to produce the 'Anatomical Deconstruction' series (see Figure 6) between 2011

and 2014. This abstracted work economised on materials and visual language. The broken edges of the smashed ceramics could be interpreted as another form of Lacan's 'rim'—they possessed mouth-like orifices. The glass casts that I created from the negative spaces inside the sink became alternative plugs.

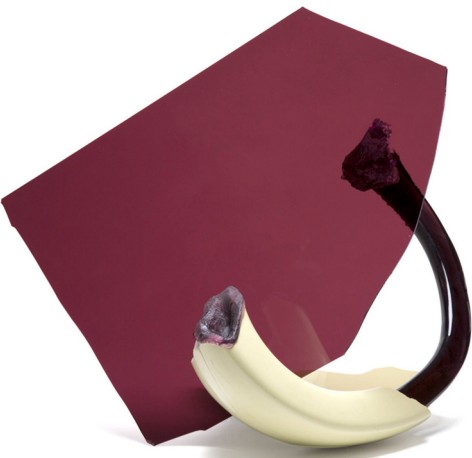

**Figure 6.** Matthias. 2012. Anatomical Deconstruction VI. 50 cm H, 40 cm L, 53 cm W. Sheet glass, cast glass and found object.

Lacan could possibly identify and 'define these peculiar objects by the formula *objet petit a*, the 'objects' of the sexual drive. The 'objet a' is part of the subject which is 'detachable' from the body and is thus capable of confronting the subject as alien and external. The 'objet a' is typically auto-erotic, a part of the subject that can be treated as if it were an external object' (Gross 1990, p. 88). According to Lacan, 'objet a' is not an object but an action, an absorption which only satiates 'the want, the absent' for a short period. It could be interpreted as a longing or a rejection of a feeling of dependency (Gross 1990, p. 88). Hidden spaces were made real and became the marrow of the bone-like receptacle, which satisfied as an 'objet a' in the same way as the sink plug. The legacy of this series of work is ever present in my current conceptual considerations and, therefore, relevant to discuss in this paper. Anatomical references prevail, but my work has materially and conceptually expanded.

*3.3. Clean-Up Workers (Deluxe Series)*

In 2021, an ongoing series of mixed media sculptures, Clean-Up Workers (Deluxe Series) I-IX, was born from the re-appropriation of the misinterpreted 'Sink and Plug'. I attacked this work with a hammer (see Figure 7). I filmed the process. It was cathartic to destroy work that I felt represented tentative first steps, even though the work was formative in directing my future conceptual and material investigations. But I had never before vandalised finished work to make new work. 'Fragments, by their definition, cannot be just or only fragments, they must carry a sense of fragmentation from something else, even if what exactly they are fragments of remains open to doubt and speculation' (Viney 2015, p. 43).

As mentioned, I had previously discovered that deliberately and aggressively breaking the complete sink basin created dynamic and intriguing integrants. Internal cavities were revealed, and lyrical shattered edges balanced the pristine glazed skin and the unglazed underbelly. Each slither of ceramic still signified the contrast between cleanliness (order) and disorder. This reminds me of Tony Cragg's 'Four Plates' (1976). Comprising one intact manufactured plate and three broken plates, formalised in a line on a concrete floor. There is a progression of fragmentation and damage portrayed with each plate, but also, the fragments within each 'plate' radiate like an expanding universe. It is laid out almost like

archaeological diagnostic fragments of potsherds that will lead to minute inspection. They also suggest the stages of the return to dirt discussed by Douglas.

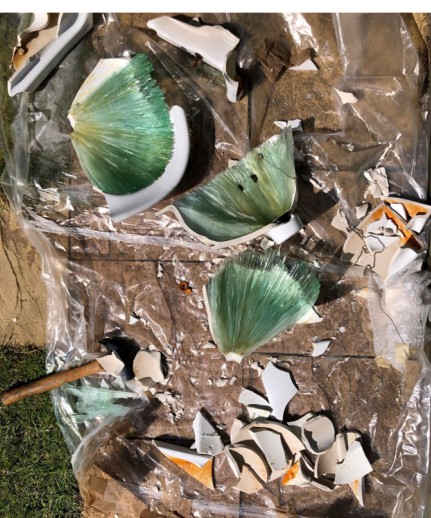

**Figure 7.** Photograph of broken '*Sink and Plug*', 2021.

The breaking of 'sink and plug' revealed sections of the constructed sheet glass that lined the ceramic basin liberated from the supporting ceramic; it was as if the sink organ was now revealing its gills. I responded to the random fragments that I had created as a means to imagine a series of anatomical cleaning devices.

The vacuum cleaner is already a visual sign bearing ergonomic proportions, and I exaggerated this by merging it with anatomical features that resemble organs and bones. I had made an earlier piece of work entitled The Os Coxa Diaphragm Chamber 142 (see Figure 8). The title of that work stemmed from my interpretation of a brand name for an 'anatomical vacuum cleaner' and featured in 'S-O-T Body Repairs' exhibition in 2016. Scanlan identifies a range of cleaning objects, including the vacuum cleaner, and states that 'these objects represent progress and significantly stand in a relation of identity to its function, which we can see if we define such objects in terms of this functionality' (Scanlan 2005, p. 110). After WWII, they were specifically marketed to housewives to offer efficient, hygienic control over domestic spaces, ''lifestyle cleaners' who view house-cleaning as the universal antidote to contemporary life's ailments' (Scanlan 2005, p. 174). Though, as my mother says, 'houses have got big mouths', so the housework and expenditures are never 'done'. In the early 20th century, it was aspirational to be a working-class stay-at-home wife, mirroring upper- and middle-class families. This is no longer the case. Roberts noted that 'Domestic ideology was such that housework became understood not just as a set of chores but as a moral undertaking' located in suburban settings (Cited in Valentine 2001, p. 67). Scanlan states that one's proximity and interaction with the handling or organisation of waste is in direct correlation with one's social status as individuals and as part of the body politic (Scanlan 2005, p. 169).

'Clean-up Workers' is a term that I encountered when researching the clean-up of radioactive waste after the accident at the Chernobyl Nuclear Power Plant in Ukraine. It seems an innocuous, child-like term for such a dangerous industrial-scale rectification that resulted in many deaths from acute radiation syndrome. I wanted to use it as a generic descriptor denoting all types of cleaners. The ironic qualifier, 'Deluxe Series', was added to mimic names such as Hoover's 'deluxe' Model 63, which debuted in 1953. In 1957, Hoover introduced the Convertible Model 65 (the De Luxe 652 in the UK). It was the last machine designed by the industrial designer Henry Dreyfuss. Deluxe was also used to describe automobiles; the Chevrolet Deluxe was marketed from 1941 to 1952.

'Deluxe' in 'Latin had two similar words with overlapping meanings: *luxus*, meaning "luxury" or "excess," and *luxuria*, which meant "rankness" or "offensiveness." These terms

became *luxe* and *luxure* in French, with meanings that preserved the distinctions of the original Latin. The French word *luxe* developed the meanings that we commonly associate with *luxury*: "wealth" and "indulgence." The French phase *de luxe*, meaning "of luxury," became the English word *deluxe*' (Merriam-Webster.com 2016).

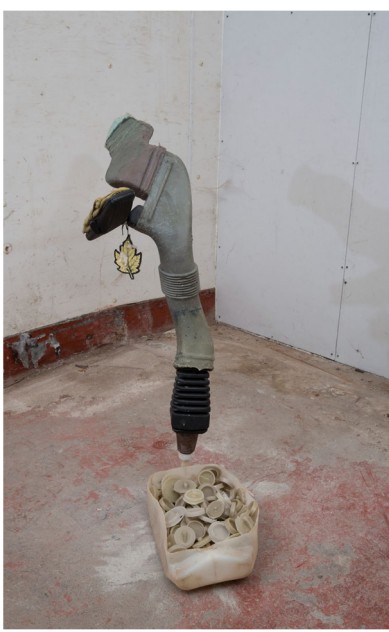

**Figure 8.** Matthias. 2016. The Os Coxa Diaphragm Chamber 142. 87 cm H, 80 cm L, 33 cm W. Wax, steel armature, found objects.

There are parallels between 'deluxe' and 'sink'. Both words contain a synthesis of desirable and undesirable peculiarities.

The ordering or removal of 'matter out of place'—Douglas's definition of dirt—is an important aspect of the employment of these domestic and industrial devices. The assignment and ordering of waste in private spaces has resulted in the identification, treatment and removal of different types of waste. I do perceive that certain waste objects and materials that I use in my work can be viewed as uncanny. Freud explains that the German translation of uncanny is 'unheimlich, the opposite of heimlich, meaning "familiar," "belonging to the home"; and therefore, what is "uncanny" is frightening because it is not known and familiar. Waste is unhomely. There is an irony in making imagined cleaners of waste from waste materials that exist outside the home and imagining that their domain is this private sphere. Freud states, 'That many things that would be bound to seem uncanny if they happened in real life are not so in the realm of fiction' (Freud 2003, p. 156). So, I will not refer to the work that I make as uncanny as it also exists in the realm of the imagination.

Referencing the anatomy of a vacuum cleaner helped to establish the key requirements to be considered in each composition: handles, tubes (some of which are translucent), flexible hoses, bristles, filters, caps, sacs or containers, scoops or pans, extension leads, and small attachments. The smaller pieces have docking stations that frame the devices. Each piece is, therefore, a combination of fragment(s) from 'Sink and Plug' with by-products, blown glass tubing, rubber and plastic hoses and car mechanical parts or deformed cast glass versions of these and other broken found objects that already have associations with the body.

In five of the pieces, I incorporated various vacuum cleaner bags/sacs; they are representative of lungs or other organs filled with pollutants. One bag was made from a worn settee leatherette cover I found by the side of the road (II), two from inverted gear stick covers I acquired from a scrapyard (III and VII), and one was cast in transparent yellow glass moulded from assembled indicator stalk rubber caps that came from the same

scrapyard. One was a small, calibrated plastic cylinder used to collect a blood sample, which was inadequate for the job (VI). This came from a scrap store.

I began the process of re-appropriation with the more substantial fragments; the first two pieces in the series (I and II) used the sink fragments that contained the now-exposed glass shards that resembled bristle-like implements or defunct filters clogged with dust and debris. If employed, they could scratch and tear any surface they came in contact with, including the user. Both compositions are deliberately awkward in assemblage, almost off balance. In exploring the 'artistic balance between order and disorder' (Scanlan 2005, p. 95), I often have to consider the weight of materials and counterbalance using metal armatures.

Clean-Up Workers I (see Figure 9) has two glass pipes; the shorter one forms a handle covered in sticky wax with a pink scourer feature, and the larger, translucent, cream 'wind tunnel ultra-grip rod' contains dust and sediment. There is a tiny toothpick/flossing head attachment (found by the side of the road) at the end of the red cable for those difficult-to-reach nooks and crannies that suggest obsessive cleaning. A ploy that I use is to contrast the intricate with larger components as this 'suggests an ambiguity of scale from the intimate to the monumental' (Haigh 2014, p. 67).

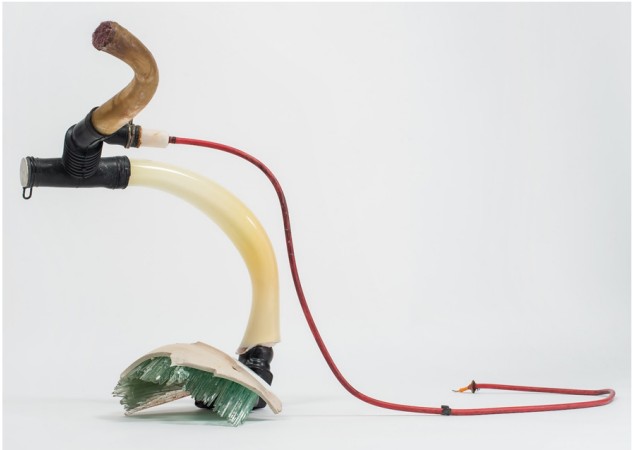

**Figure 9.** Matthias. 2021. Clean-Up Workers I (Deluxe Series). 63 cm H, 161 cm L, 30 cm W. Blown glass, found objects, sheet glass and wax.

Clean-up Workers II (see Figure 10) has a small plughole cover attached to a chain that anchors the bag structure to the wall, a nod to the sink plughole, either restraining or empowering the movement of the cleaner. The dust bag is attached to a 'flexible hose', which is attached to a 'bellows dirt cup'. The anatomy of a vacuum cleaner is intriguing, especially the names given to parts. It is probably a good thing that I do not know what the car parts are, as I might then attach a monetary value to them. I chose the parts because of their association with the corporeal through form and material analogy. This piece was acquired by the European Museum for Modern Glass, Coburg, in 2022 to be part of their permanent collection.

I reclassified the larger 'uncontaminated' broken ceramic sink fragments as scoops or pans. These can be found in pieces III and IV. Clean-Up Workers III (Deluxe Series) (see Figure 11) is a versatile piece of kit as it has both a (nail) brush feature at one end and it can collect debris in the inverted gear stick bag at the other end. The phallic gear stick is cradled inside the sac so it cannot hold much waste. There is plastic swarf plugging the bag (previously generated waste). The device is reminiscent of a flying insect catcher. However, it is frail as it has a cast glass umbilical cord that unites the two utensils. The cast was taken from a cavity within the ceramic fragment that was then moulded and replicated a number of times. It also serves as an 'objet petit a' as it slots and fills the cavity but, when rested, hovers precariously above the work surface.

Clean-Up Workers IV (Deluxe Series) (see Figure 12) is a top-heavy precarious scoop that is anchored to a length of rusty steel that provides an impractical housing mechanism.

Again, the 'wind tunnel ultra-grip rod' is translucent and full of waste, but it is unclear as to how the waste got there. This device has a sponge attachment. It is a transitional bridge between the first three pieces and the latter ones that have framing devices. Unfortunately, there is no brush to accompany the pan, so it is somewhat redundant, a vacuum cleaner with a planned obsolescence.

I then progressed to use smaller, more insignificant shards of ceramic sink for Clean-Up Workers (Deluxe Series) V-IX in order to make more intimate pieces.

The central component, the cleaning devices, are staged and framed to contextualise their meaning and create a window into another world, a mise en scene, or 'a bounded space of representation' (Scanlan 2005, p. 92). Each tableau that supports the Clean-Up Workers is disintegrating, beyond repair and beyond surface cleaning or treatment. They operate on a micro-macro level, and they are internal and external. They also provide a rim and an 'objet petit a' that cannot suffice. Cleaning will never be fulfilling as it never ends.

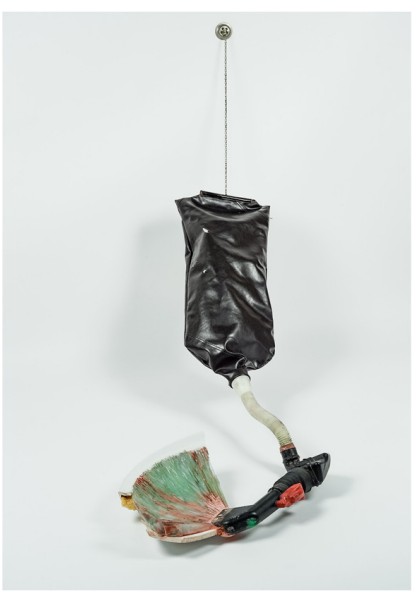

**Figure 10.** Matthias. 2021. Clean-Up Workers II (Deluxe Series). 125 cm H, 132 cm L, 72 cm W. Leatherette, plug hole cover, sheet glass and found objects.

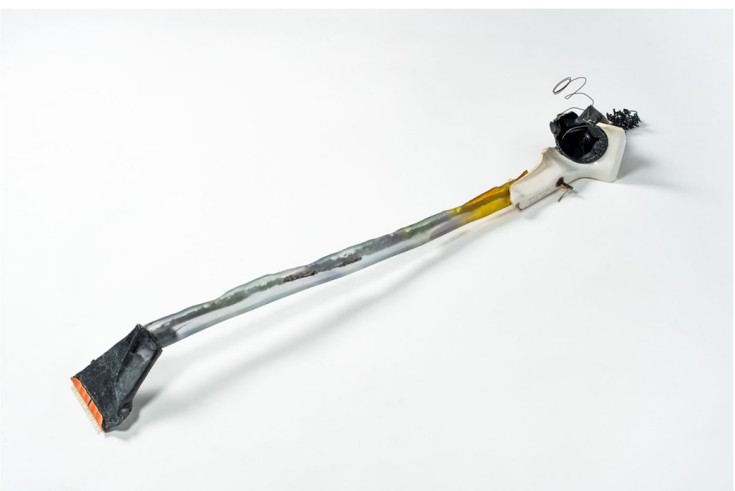

**Figure 11.** Matthias. 2022. Clean-Up Workers III (Deluxe Series). 13 cm H, 136 cm L, 21 cm W. Cast glass, found objects, steel strap and swarf.

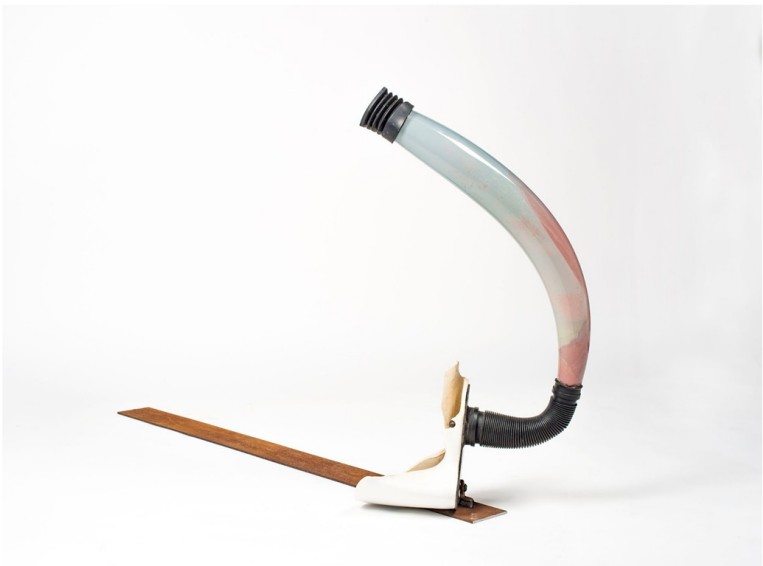

**Figure 12.** Matthias. 2022. Clean-Up Workers IV (Deluxe Series). 82 cm H, 128 cm L, 22 cm W. Blown glass, sheet steel and found objects.

In Clean-Up Worker VII (Deluxe Series) (see Figure 13), the red car filter takes on the role of an architecturally moulded picture frame. It also provides a fleshy, plush, undulated landscape for the urine-coloured cast glass organ to rest on (a nod to the piss bottles). The cabling intrudes beyond the boundary of the frame in a similar way to the trompe l'oeil painting of 19th-century realist painter Pere Borrell del Caso's most famous work, Escaping Criticism (1874). The cabling is held together by a cast bronze replica of a plastic toy arm that I found on a beach that has a threatening or defiant fist. It speaks of the threat of 'you must clean your room' or 'I shall not clean my room, what a waste of time'.

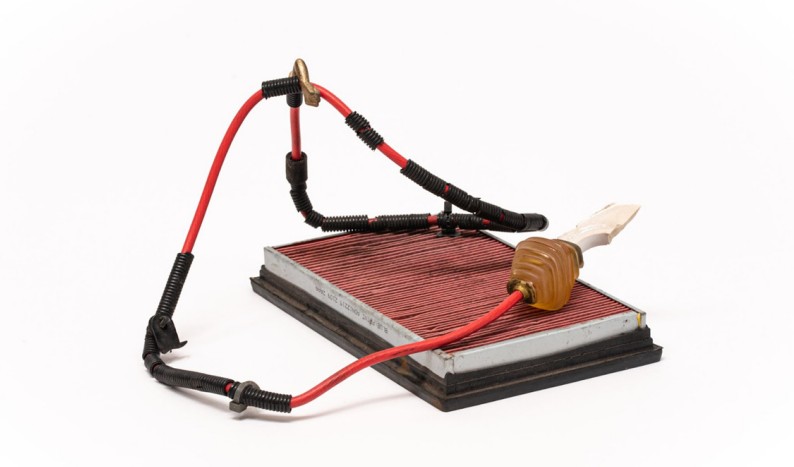

**Figure 13.** Matthias. 2022. Clean-Up Workers VII (Deluxe Series). 21 cm H, 32 cm L, 29 cm W. Cable, cast metal, cast glass and found objects.

In Clean-Up Workers VIII (Deluxe Series) (see Figure 14), the ceramic shard/scoop also breaks from the metal circle and interrupts repetitive loops, the dominant feature of the composition. The metal circle also frames and protects the inadequate cylinder that is used as a depository for dirt and dust that is skin and hair—the debris from vacuuming.

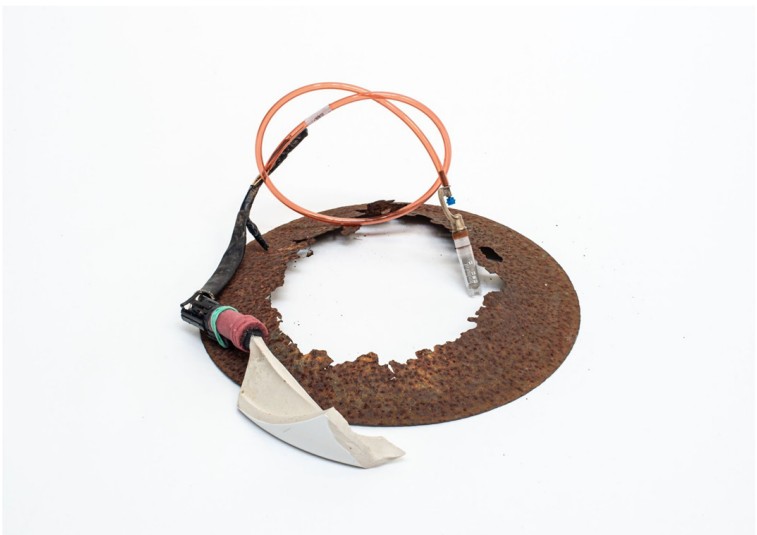

**Figure 14.** Matthias. 2022. Clean-Up Workers VIII (Deluxe Series). H 10 cm H, 44 cm L, 30 cm W. Rusty metal ring, cast glass, found objects and dust.

Clean-Up Workers V (Deluxe Series) (see Figure 15) is currently an unresolved piece. Originally, the cleaner was supposed to hang from a stand/docking station, but on photographing the work, I discovered the anthropomorphic nature of the composition was dominant, so the piece now lies on its side; there is a suggestive eye latch feature. V also has a gear stick cover sac filled with debris, which is attached to respirator packaging that looks like a urinal but also suggests the inhalation of dust if fitted over the mouth. The ceramic cavity is filled with a sponge.

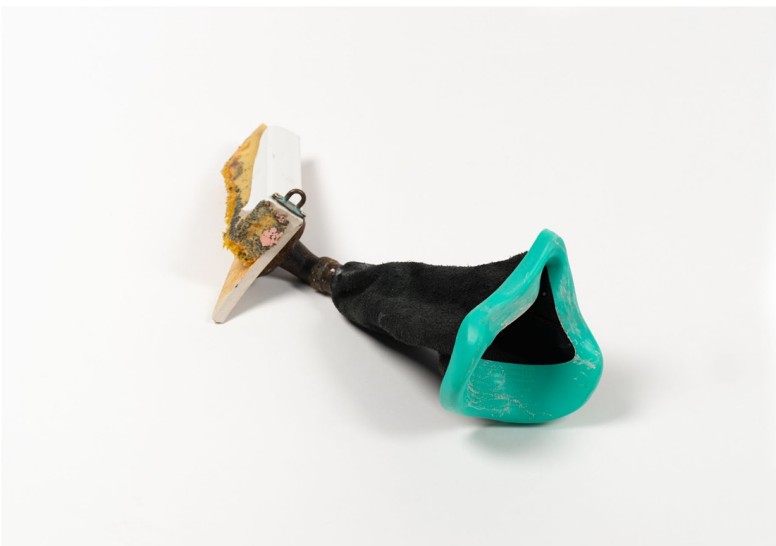

**Figure 15.** Matthias. 2022. Clean-Up Workers V (Deluxe Series). 14 cm H, 39 cm L, 20 cm W. Fabric, cast glass, found objects and Jesmonite.

Clean-Up Workers VIII (Deluxe Series) (see Figure 16) sits almost like a dollhouse vacuum cleaner on a crumpled car registration number plate rug. I used waste plaster in combination with cast glass. It suggests that it is spewing from its container, and the vacuum cleaner again is defunct. This piece, along with Clean-Up Workers VII and IX, was selected to be included in the Ireland Glass Biennale 2023.

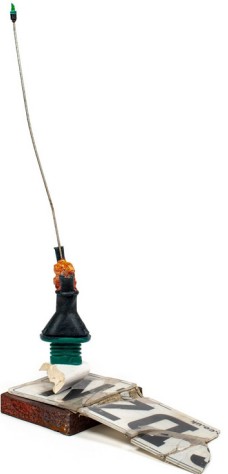

**Figure 16.** Matthias. 2023. Clean-Up Workers VIII (Deluxe Series). 48 cm H, 28 cm L, 13 cm W. Number plate, metal frame, cast glass, spring, and Jesmonite.

For Clean-Up Workers IX (Deluxe Series) (see Figure 17), I used a vulnerable protrusion of the car, the discarded wing mirror, which is an indicator of some kind of collision, a misjudgement by the driver. It is an abandoned object; smashed mirrors tend to be located by roadsides, particularly on narrow lanes, and they can be difficult to find. The mirror is also of interest as it allows the driver to see outside of peripheral vision to blind spots. I was reading about Lacan's notions of the mirror phase in a child's psychological development, the point at which the child first recognises a distinction between itself and the (m)other, the point at which it also begins an understanding of space, distance and position. The wing mirror is almost a reverse textural composition to the cushioned dirty scourer that it sits upon, though there is something glitzy about the mirror ball effect that I find captivating; it elevates the scoop on a disco podium. The scoop contains the debris from cutting and filing the black plastic back of the mirror in order to engineer the top-heavy composition. The opaque lilac cast glass mimics plastic or innards. It has a mouth-like opening that is lined with silver wire, out of which spews the waste. This waste is another form of 'objet petit a', analogous to consumerism, that momentary satisfaction through acquisition or expulsion.

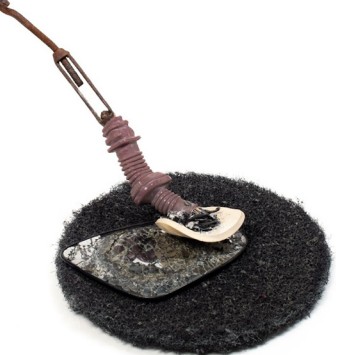

**Figure 17.** Matthias. 2023. Clean-Up Workers IX (Deluxe Series). 26 cm H, 38 cm L, 30 cm W. Found objects, silver, cast glass and broken mirror.

All is not well in the domesticated space.

## 4. Conclusions

'Narrative contexts are so often evoked by waste things' (Viney 2015, p. 44). Clean-Up Workers (Deluxe Series) demonstrates that by combining idiosyncratic waste selections of 'by-products of my practice' with the 'owned objects or materials' alongside bespoke crafted objects, the narratives of value and abjection become inherent in the work. I have

extended the use-time of the once-defunct objects and materials, giving them a second life, a form of redemption. The rarity of finding certain types of waste, like the smashed wing mirror in non-spaces, such as the gutter, elevates the status of these objects in my mind. But this can also be achieved by its 'position within a context'. 'Objects gain their value through the situations in which they are placed' (Johansson 2012, p. 9). Through material or object associations, in this instance, high-value kiln-formed glass meets low-value quotidian waste. But also, by exhibiting my assemblages in museums and galleries, I am recontextualising that waste into something special, a one-off artefact that contains symbolic meaning. 'Sink and Plug', through its re-appropriation into Clean-Up Workers (Deluxe Series), died a violent death but was resurrected to become a more succinct and relevant body of work that has so far featured in two major international exhibitions.

In order to manage and provide a counterbalance to the simmering themes of death, residue and a life less lived, I inject absurdity into my work through the use of titles and descriptions of the imagined world that I see my anatomical vacuum cleaners operating in. I reference the isolation, tedium and oppression of the female silent generation that was supposedly satiated by the purchase of domestic commodities. I perceive my 'implements of order' have also reached a point of exhaustion.

Discussing ideas about bodily waste and how it is managed in private spheres mirrors Western societies' attitudes and attempts to conceal or deny waste, be it bodily or otherwise. The philosophy behind waste in Western cultures appears to suggest a malaise that needs treating. The 'objet petit a' feature consistently as a 'plug' in my work, but it can be interpreted in many ways from a phallic symbol, a pacifier, food or cigarettes, or it could represent the general desire to consume—to satisfy a need that can never fully be met in order to try to obliterate the other. It is associated with corporeal margins, and these margins of social, cultural and political ideologies are vulnerable. It is also associated with how a society responds to the body, and these margins mirror societal structures and identities, as Douglas identifiesAs Thill observes, 'In rejecting our waste, we seek to reject everything associated with it' (Thill 2015, p. 44), and this includes ourselves. Although my work is autobiographical in nature, waste is the common denominator of all existence, and what we do not value and choose to discard should not be brushed under the carpet, particularly as a topic of discourse within the world of glass art. Perhaps more (glass) artists could develop a scavenger mentality.

**Funding:** This research received no external funding.

**Data Availability Statement:** Not applicable.

**Conflicts of Interest:** The author declares no conflict of interest.

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
