# Peer review of "Clean-Up Workers (Deluxe Series): The Embodiment of Waste Values and Aesthetics"

_arts, 2022_

Round 1

Reviewer 1 Report

Overall this was a well written article with clear outline and situates a contemporary sculptural glass practice it within the broader field of visual culture. 

The work was well situated within both historical (20th c) and contemporary arts practice. In addition its location within conceptual and theoretical understandings of waste was well placed. The discussion of the use of glass within the context of waste and bodily fluids was well placed with contemporary art practice (Dion, Hatoum and others). It would have been good to see a little more contextualisation of this work in relation to the contemporary glass scene and how this relates to the current 'state' of the field as the references in relation to this are perhaps a little dated now (e.g. Cummings, 1997)? Might it be good to see it in context of for example Emma Woffenden's work or others who might be operating or 'located on the periphery of glass art' as you beautifully put it but explain why you think this is so? You rallying call at the end for other glass artists to develop a scavenger mentality is a timely one and one which perhaps deserved more attention in the body of the text?

Overall, a good article that contributes towards writing on contemporary glass sculptural practices in particular (about which too little gets written about!) and sculpture in general.

There are some minor details which could do with clarification or extra signposting to help the reading better understand or place the work? Below are some minor editorial details which could help improve the text. Please note these are minor and easily modified.

Some of the Headings are perhaps slightly misleading: e.g. 2. Results feels more like the part of the introduction than the results per se?

It would be good to perhaps see a few details clarified, mainly around the introduction of references and contextualisation to ease the reader's journey.

For example: 

line 169: who is Kristeva: add a few word to enable placement or context

line 214/line 227 : similar; e.g. add 'artist' or other relevant word to locate Cornell and Dion respectively

line 195: who is Viney and from what field or knowledge field is he writing? 

line 222: the Irigaray quote needs a short joining statement linking it with the previous sentence; who is Irigaray writing about? 

line 270, Whalley and Miller - who are they? Situate them

line 272: Rick Adam: who is he: an artist? photographer? Short introductory work would be helpful

line 299: Gross: who is he/what field?

line 189: is missing a word perhaps to make it full sentence? 

Overall I enjoyed reading this article and think it will set well within the Special Issue on Art Glass Studies for a Changing World – the International Year of Glass

Author Response

Thank you for the constructive feedback. 

I have addressed all minor modifications that you have outlined including the introduction of references and contextualisation of these.  

I have included Emma Woffenden's work as an example of contemporary glass practice that has some parallels with my own practice in terms of methodology and material usage.  

I have slightly reworded the Results section of the paper.

I recognise that my advocacy of a scavenger mentality is a 'timely and one which perhaps deserved more attention in the body of the text' but without changing the balance of the paper I am not sure how to include this more fully?

I am pleased that you consider this paper to make a good contribution to contemporary glass sculptural practices and sculpture in general.

Reviewer 2 Report

n/a

Author Response

Thank you for the feedback.

Reviewer 3 Report

The article is very well organized and demonstrates recent work. The approach to glass is different from what we have observed, contributing for the enrichment of glass art

Some deformation on the references, please cheek and some aren't in alphabetic order

Line 34 – the author says – “I have been incorporating waste objects and materials in my work since 2007 and am one of a small number of artists working with glass that …” who are the other artists?

Line 122 – “The Romans reined and capitalised on glassblowing. It became transformational and ubiquitous because of their efficient, large scale technological advancements that resulted in a long hiatus in kiln-formed glass production. “ – this is not clear, can you please explain . How it resulted in a long hiatus in kiln-formed glass production?

Line  184  - the author speaks about Cornelia Parker – it is important, but the paragraph is a little confuse

Figure 3 and 4– can you please provide the place

Line 290 - can the author explain  what is ‘ Abject Art'

Line 549  - when was the piece acquired by the museum?

Figures legends of the author work are incomplete. It is important to have sizes to understand the scale. Also, it would help the reader if the materials used are mentioned 

Author Response

Thank you for the constructive feedback.

I have made the minor recommendations that you have highlighted to both the text and labelling of images to better inform the reader.

I am pleased that you found the article to be well organised and informative of current creative practice.